# Impact of COVID-19 on Out-of-Hospital Cardiac Arrest in Singapore

**DOI:** 10.3390/ijerph18073646

**Published:** 2021-03-31

**Authors:** Shir Lynn Lim, Nur Shahidah, Seyed Ehsan Saffari, Qin Xiang Ng, Andrew Fu Wah Ho, Benjamin Sieu-Hon Leong, Shalini Arulanandam, Fahad Javaid Siddiqui, Marcus Eng Hock Ong

**Affiliations:** 1Department of Cardiology, National University Heart Center, Singapore 119228, Singapore; 2Department of Emergency Medicine, Singapore General Hospital, Singapore 169608, Singapore; nur.shahidah.ahmad@singhealth.com.sg (N.S.); sophronesis@gmail.com (A.F.W.H.); marcus.ong.e.h@singhealth.com.sg (M.E.H.O.); 3Center for Quantitative Medicine, Duke-National University of Singapore Medical School, Singapore 169857, Singapore; ehsan.saffari@duke-nus.edu.sg; 4Emergency Medical Services Department, Singapore Civil Defence Force, Singapore 408827, Singapore; ng_qin_xiang@scdf.gov.sg (Q.X.N.); shalini_arulanandam@scdf.gov.sg (S.A.); 5SingHealth Emergency Medicine Academic Clinical Programme, Duke-National University of Singapore Medical School, Singapore 169857, Singapore; 6Pre-Hospital and Emergency Research Center, Duke-National University of Singapore Medical School, Singapore 169857, Singapore; fahad.siddiqui@duke-nus.edu.sg; 7Emergency Medicine Department, National University Hospital, Singapore 119085, Singapore; benjamin_sh_leong@nuhs.edu.sg; 8Health Services and Systems Research, Duke-National University of Singapore Medical School, Singapore 169857, Singapore

**Keywords:** coronavirus disease 2019, out-of-hospital cardiac arrest, return of spontaneous circulation

## Abstract

This study aimed to evaluate the impact of the Coronavirus Disease 2019 (COVID-19) pandemic on out-of-hospital cardiac arrest (OHCA) in Singapore. We used data from the Singapore Civil Defence Force to compare the incidence, characteristics and outcomes of all Emergency Medical Services (EMS)-attended adult OHCA during the pandemic (January–May 2020) and pre-pandemic (January–May 2018 and 2019) periods. Pre-hospital return of spontaneous circulation (ROSC) was the primary outcome. Binary logistic regression was used to calculate the adjusted odds ratios (aOR) for the characteristics of OHCA. Of the 3893 OHCA patients (median age 72 years, 63.7% males), 1400 occurred during the pandemic period and 2493 during the pre-pandemic period. Compared with the pre-pandemic period, OHCAs during the pandemic period more likely occurred at home (aOR: 1.48; 95% CI: 1.24–1.75) and were witnessed (aOR: 1.71; 95% CI: 1.49–1.97). They received less bystander CPR (aOR: 0.70; 95% CI: 0.61–0.81) despite 65% of witnessed arrests by a family member, and waited longer for EMS (OR ≥ 10 min: 1.71, 95% CI 1.46–2.00). Pre-hospital ROSC was less likely during the pandemic period (aOR: 0.67; 95% CI: 0.53–0.84). The pandemic saw increased OHCA incidence and worse outcomes in Singapore, likely indirect effects of COVID-19.

## 1. Introduction

Coronavirus Disease 2019 (COVID-19) has resulted in unprecedented health, economic, and social consequences. Officially declared a pandemic by the World Health Organization (WHO) on 11 March 2020 [1], it has since spread to 219 countries as of 31st December 2020, exceeding 82 million infections and 1.8 million deaths globally [2].

Public health efforts to reduce transmissions in afflicted countries have included changes in healthcare provision and delivery, as well as compulsory confinement and restriction of movement of people, which may result in systematic delays and negatively influence health-seeking behavior. Paradoxical declines in hospitalizations for acute cardiovascular illness have been reported, accompanied by delayed presentations and higher rates of in-hospital mortality [3,4,5,6]. These highlight the collateral consequences of the pandemic on healthcare systems, beyond the direct mortality and morbidity caused by COVID-19.

Management of out-of-hospital cardiac arrest (OHCA) requires prompt and coordinated efforts from laypersons, Emergency Medical Services (EMS) and hospital providers. Its outcomes reflect the population health, efficacy of the healthcare system and health-providing behavior during the pandemic. Marked increases in OHCA incidence with worse outcomes during the pandemic have been reported in Greater Paris, New York City and Lombardy (Italy), areas badly afflicted with COVID-19 [7,8,9]. In contrast, regions with low prevalence rates of COVID-19 such as Victoria (Australia) and Pennsylvania (USA) did not observe increases in OHCA incidence [10,11]. Little is known about the impact of COVID-19 on OHCA in Asia and at a national level in Singapore.

Using preliminary data from Singapore’s national ambulance service, the Singapore Civil Defence Force (SCDF) [12], we aimed to compare the incidence and outcomes of OHCA during the COVID-19 pandemic with pre-pandemic period, hypothesizing that the COVID-19 pandemic would affect the incidence, characteristics and health-provision of OHCA, leading to changes in OHCA outcomes.

## 2. Methods

This retrospective cohort study is reported according to the STROBE (strengthening the reporting of observational studies in epidemiology) guidelines [13].

### 2.1. Study Design and Setting

This nationwide cohort study included adult (18 years or older), EMS-attended OHCA of all etiologies occurring in Singapore between 1 January and 31 May in the years of 2018, 2019 and 2020.

Singapore is a densely populated, multi-ethnic city-state in the Asia–Pacific, with a population of 5.7 million over a land area of 725.7 square kilometers (km^2^), giving a population density of 7866 persons per km [2,14]. Over 90% of the population resides in high-rise apartments. The COVID-19 pandemic in Singapore is summarized in Figure 1. It was one of the first few countries to be affected by COVID-19 after China, documenting its first case on 23 January 2020 [15]. Its Disease Outbreak Response System Condition alert was raised to “orange”, the second highest level, on 7 February 2020 in response to local COVID-19 transmissions [16]. Increasing numbers of new infections from 6 March 2020 onwards necessitated stringent public health measures and travel advisories, culminating in a partial national lockdown termed the “Circuit Breaker”, enforced from 3 April 2020 to 2 June 2020 [17]. Hospitals temporarily halted noncritical services; other measures included school closures, a mask advisory, safe distancing regulations, and closures of beaches and playgrounds. Outbreaks of COVID-19 in the foreign workers’ dormitories started in April 2020, which led to exponential increases in daily new cases and total infections. As of 31 May 2020, there were 34,884 cases with 23 deaths (prevalence rate of 612 per 100,000 population and case fatality rate of 0.07%) [18].

The SCDF provides nationwide EMS in Singapore. It is a fire-based system activated by a centralized “995” dispatch system, responding to more than 190,000 calls each year. It has a fleet of 84 ambulances, and all are equipped with mechanical cardiopulmonary resuscitation (CPR) devices. A protocol implemented in 2012 saw trained emergency call-takers provide CPR instructions to bystanders. Each OHCA case is attended by an SCDF ambulance comprising a paramedic and two Emergency Medical Technicians (EMT), with one as the ambulance driver. Where necessary, motorcycle-based EMTs or firebikers are dispatched ahead of ambulances. Termination of resuscitation (TOR) protocol for OHCA was introduced in January 2019, and a tiered response to OHCA was implemented in April 2019 [19]. As part of the tiered response, additional fire medical vehicles are deployed for enhanced medical support; the presence of additional EMTs allows for high-performance CPR. In response to the COVID-19 pandemic, the SCDF reverted to a single-tiered response to OHCA from 7 February 2020; fast-response bikes and fire appliances stopped being deployed, and high-performance CPR was discontinued. Paramedics and EMTs were required to don full personal protective equipment (PPE) for all cases attended.

### 2.2. Data Sources

Data for this study were imported from the SCDF EMS data repository. These EMS records were collected via data input by trained paramedics responding to EMS calls and maintained by an internal audit team. Data for the first five months of the years 2018, 2019 and 2020 were analyzed.

### 2.3. Data Elements and Definitions

All data definitions for OHCA adopted by the SCDF EMS data repository are in accordance with Utstein definitions [20]. Cardiac arrest is defined as the cessation of cardiac mechanical activity confirmed by the absence of signs of circulation at any time as documented on the EMS treatment record. The EMS-attended population represents all OHCA cases attended by the EMS and includes both cases who receive emergency treatment and those who are declared deceased on EMS arrival. Bystander CPR is defined as any attempt at chest compressions, with or without ventilation, and is assumed to be absent if not stated. Bystander automated external defibrillator (AED) application is defined as the deployment of public AED, regardless of whether it delivered a shock, and is assumed to be absent if not stated. The total response time (in minutes) is the interval between time of call received by the dispatch center and the time of patient contact by either the ambulance or rapid responder dispatched via the same dispatch center. Pre-hospital return of spontaneous circulation (ROSC) refers to the resumption of perfusing cardiac activity either at the scene or during transport, whether sustained or not.

The COVID-19 pandemic period refers to the period between 1 January 2020 to 31 May 2020, and pre-pandemic period refers to 1 January to 31 May 2018 and 2019.

### 2.4. Statistical Analysis

Demographics and baseline characteristics of adult EMS-attended OHCA patients were reported for January–May 2018, 2019 and 2020 as median (first to third quartile—Q1–Q3) and frequency (percent) for continuous and categorical variables, respectively. OHCA characteristics were compared between the pandemic period and pre-pandemic period using binary logistic regression analysis. Multivariable logistic regression procedure was performed to investigate the differences in OHCA characteristics between the two periods after adjusting for potential confounders, which were chosen based on statistical significance and clinical relevance. These included age, gender, first rhythm of arrest, location type, witnessed arrest, bystander CPR performed and bystander AED applied; age was treated as a continuous variable while the others were treated as categorical variables. The same methodology was used to compare the odds of clinical outcome (pre-hospital ROSC) between the two time periods while adjusting for age, gender, location type, witnessed arrest, bystander interventions, first rhythm of arrest, pre-hospital defibrillation and total response time, with age and total response time treated as continuous variables and other remaining variables as categorical. Significance level was set at *p*-value of <0.2 at variable selection stage and <0.05 at model development stage. Odds ratios (OR) of observing a characteristic or an outcome between the two periods were calculated and interpreted. Meaningful interactions were also evaluated. Model adequacy was assessed by Hosemer and Lameshow Goodness of fit test. Statistical analyses were performed using SAS software version 9.4 for Windows (SAS Institute Inc: Carry, NA, USA).

## 3. Results

### 3.1. Overall Characteristics

From 1 January to 31 May in the years 2018 to 2020, there were 3893 cases of adult OHCA (Figure 2), with a median (Q1–Q3) age of 72 (60–83) years, and of whom, 2479 (63.7%) were males and 2701 (69.4%) were Chinese. Of all the OHCAs, 2107 (54.1%) were witnessed arrests, 2890 (74.2%) occurred in home residences, and 595 (15.3%) had initial shockable rhythms. While 2248 (57.7%) received bystander CPR, only 339 (8.7%) had bystander AED applied. The median (Q1–Q3) total response time was 11.9 (9.9–14.6) minutes; resuscitation was attempted in 3838 (98.6%), 3623 (93.1%) were transported, and pre-hospital ROSC was achieved in 455 (11.7%).

### 3.2. Baseline Characteristics

The characteristics of EMS-attended adult OHCA population are summarized, by year, in Table 1. OHCA at home accounted for the majority of cases in all 3 years, with 2020 reporting the highest proportion of witnessed arrests (61.9% vs. 53.6% and 46.1% in 2018 and 2019, respectively). Bystander CPR was lowest in 2020 (52.1% vs. 61.6% and 60.3% in 2018 and 2019, respectively); while unassisted CPR was similar these three years (19.4% in 2020 vs. 18.7% and 18.6% in 2018 and 2019, respectively), dispatch-assisted CPR was lowest in 2020 (32.6% vs. 42.9% and 41.7% in 2018 and 2019, respectively). Bystander AED application increased from 2018 to 2019 (5.4% to 11.1%) and declined in 2020 (9.4%).

Total response time was similar in 2018 and 2019 (median 11.9 min and 11.3 min, respectively) and longer in 2020 (median 12.6 min). Time spent at scene was longest in 2020 (median 24.3 min vs. 22.4 min and 22.7 min in 2018 and 2019, respectively).

### 3.3. Temporal Trends

There was an upward trend in the absolute numbers and incidence of adult OHCA in Singapore over the three years (Figure 3). Pre-hospital ROSC rates were similar in 2018 and 2019 (13.5% and 12.5%, respectively), but saw a decline in 2020 (9.4%).

### 3.4. Comparison between Pandemic and Pre-Pandemic Periods

While patient demographics did not vary significantly between the pandemic and pre-pandemic periods, we observed significant differences in event characteristics, care provision and patient outcomes between the two periods (Table 2). Compared with the pre-pandemic period, OHCAs during the pandemic period had higher odds of occurring at home (OR: 1.28; 95% CI: 1.10–1.49) and being witnessed (OR: 1.64; 95% CI: 1.44–1.88). Yet, the odds of receiving bystander CPR were lower (OR: 0.70; 95% CI: 0.61–0.80); there was no significant difference observed in bystander AED application (OR: 1.13; 95% CI: 0.90–1.43). The total response time was longer during the pandemic period (OR ≥10 min: 1.71; 95% CI: 1.46–2.00)—this was contributed by longer time from dispatch to scene arrival (OR of ≥6 min: 1.17; 95% CI: 1.02–1.33) and time from scene arrival to first patient contact (OR of ≥2 min: 1.41; 95% CI: 1.20–1.66). The odds of experiencing pre-hospital ROSC were lower during the pandemic period compared to pre-pandemic period (OR: 0.69; 95% CI: 0.56–0.86).

These differences in OHCA characteristics persisted in subsequent analyses with logistic regression (Table 3). Adjusting for clinical, circumstantial and interventional characteristics of an OHCA patient, odds of experiencing pre-hospital ROSC were lower during the pandemic period (aOR: 0.67, 95% CI 0.53–0.84) compared with the pre-pandemic period. Clinically relevant two-way interaction terms were tested to investigate whether differences between pandemic and pre-pandemic periods depend on the level of a third variable. None of the interaction terms were statistically significant.

## 4. Discussion

Singapore saw an increase in adult OHCAs during the COVID-19 pandemic, with more arrests occurring at home. Despite an increase in proportion of witnessed arrests, there was less bystander CPR. Outcomes were worse compared to pre-pandemic period, with a decline in the likelihood of pre-hospital ROSC. Our findings extended that from Europe and New York City [7,8,9], by providing the first Asian country-level report on the impact of COVID-19 on OHCA.

The increase in adult OHCA observed in 2020 was generally aligned with the secular trends in adult OHCA [21], contributed by ageing population and rising cardiovascular burden in Singapore [22]. Despite that, we could not discount the indirect contribution from the COVID-19 pandemic, due to changes in healthcare systems and health-seeking behaviour of the population. Delayed, and often sicker, presentations of acute cardiovascular conditions, such as acute coronary syndromes, have been reported worldwide [3,4,5]. It is conceivable that some of these additional OHCA cases were collateral consequences of the pandemic. OHCA as a direct result of COVID-19-related acute respiratory deterioration or cardiovascular complications are plausible [23,24], though less likely in Singapore, where overall rates of severe COVID-19 infection remained low.

Movement restrictions during the pandemic coupled with the unique intergenerational living arrangements prevalent in Singapore resulted in an increase in the proportion of witnessed OHCA occurring at home, especially that witnessed by family members. Paradoxically, we observed declines in bystander CPR during the same period; this reduction was seen amongst related and nonrelated bystanders, and exclusively in dispatch-assisted CPR. We offer a few possible explanations for these observations: (1) The global focus on COVID-19 may have adversely affected bystanders’ willingness to perform CPR, due to fear of aerosol generation from chest compression with potential transmission of COVID-19 from patient to layperson, even without mouth-to-mouth ventilation [25,26]; (2) This fear of disease transmission may be more prevalent in bystanders who were not previously trained or confident in performing bystander CPR, explaining the decrease in dispatch-assisted CPR; (3) The temporary discontinuation of community first-responder schemes, previously shown to improve community response to OHCA [27], resulted in a reduction in the pool of trained and willing laypersons for bystander interventions during the study period; and (4) Family members witnessing the arrests may not perform CPR due to other knowledge, cultural and psychological barriers, for example, fear of rib fractures or aggravating their loved ones’ condition. Similarly, we saw a decrease in bystander AED application rates, from 11.1% in 2019 to 9.4% in 2020, underscoring the fear of potential transmission of COVID-19 in the community. It is noteworthy that the decrease in bystander CPR and AED application rates during the pandemic occurred against a backdrop of ongoing national efforts to improve community interventions for OHCA, through public education, community CPR+AED training programs, community first-responder schemes and initiatives to improve availability and accessibility to AEDs [28,29]. These efforts saw improvements in community interventions for OHCA over the years, contributing to better patient outcomes [21,29]. There may be a risk of the COVID-19 pandemic rolling back these improvements.

The COVID-19 pandemic necessitated changes in EMS workflows, in order to reduce potential cross-contamination and to conserve personal protective equipment (PPE). These changes, together with increased call volumes and a higher proportion of OHCAs at home, contributed to longer times from dispatch to scene arrival and from scene arrival to first patient contact, culminating in longer total response times during the pandemic. The time taken from scene arrival to first patient contact is particularly relevant in Singapore, where a high proportion of OHCAs occur in high-rise, densely populated apartment buildings and significant challenges in vertical access are encountered. Operating in full PPE would also increase the fatigue levels of paramedics and EMTs.

We reported worse pre-hospital ROSC rates during the pandemic, a finding which remained significant after accounting for the confounding factors. This finding could be explained by the following: (1) A lower proportion of OHCA patients received bystander interventions, particularly CPR; (2) Changes in EMS workflows during the pandemic, leading to longer response times and discontinuation of high-performance CPR by first responders; and (3) OHCA patients may have been sicker during the pandemic, particularly OHCA related to advanced cardiac injury (delayed presentation of acute coronary syndrome and acute heart failure). Our findings of worse outcomes during the pandemic generally corroborated with that from Europe, New York City and Victoria, though the outcomes used in each study varied [7,8,9,10]. The primary outcome in our study, pre-hospital ROSC, accounted for ROSC that was transient or sustained until ED arrival. As such, the rates of survival to hospital admission and subsequent survival to hospital discharge were expected to be lower. Hospital data on COVID-19 status, etiology of arrest and overall outcomes of these patients would have been helpful, but were unavailable for the study.

The strengths of our study include the capture of all EMS-attended OHCA cases with uniform data collection based on Utstein definitions for reporting cardiac arrest. The data repository is maintained by an internal audit team, thereby ensuring data quality and integrity.

Our study should be interpreted in the context of the following limitations. The observational nature of the study rendered it susceptible to confounding factors not captured in the data repository. As we collected mainly essential pre-hospital data variables, we lacked information on comorbidities, socioeconomic factors and hospital-based management, and hence those cannot be controlled for. Patient comorbidities and socioeconomic factors have been shown to be important predictors of OHCA survival. Similarly, disparities in post-resuscitation care may explain the changes in outcomes; knowledge of how the COVID-19 pandemic affected post-resuscitation care would have been useful but was not available for this study. Given the time-sensitive nature of this study, we lack data on etiology of arrest and overall outcomes. Collection of hospital data and survival outcomes, which is more time-consuming, is ongoing as part of a planned follow-up study. There were varying amounts of missing data for all OHCA cases, albeit a small proportion (<2%). Finally, as with all epidemiological studies, data integrity, validity, ascertainment bias and misclassifications were potential limitations.

The narrative of COVID-19 in Singapore during the period of study was largely dominated by outbreaks in foreign workers’ dormitories involving young, relatively healthy men who were asymptomatic or mildly symptomatic [30]. This translated to one of the lowest case-fatality rates in the world. There were also measures implemented to prevent overwhelming the existing healthcare system during the pandemic—there was a separate call center (operated by the centralized “993” dispatch system) with its fleet of ambulances dedicated to ferry non-emergency COVID-19 suspects to hospitals, and purpose-designed community isolation facilities to manage mild cases of COVID-19. However, we still observed disruptions and delays in health-provision for OHCA at the pre-hospital level, in terms of community response as well as EMS systems-of-care, likely contributing to worse outcomes. This disruption in chain-of-survival of OHCA in the pre-hospital setting deserves further attention—public health interventions may be required to rebuild public confidence in responding to OHCA and EMS workflows may need to be re-evaluated.

## 5. Conclusions

The COVID-19 pandemic period in Singapore saw an increase in OHCA incidence, coupled with a reduction in pre-hospital ROSC. Indirect effects of the pandemic, resulting in delays or disruptions in community response to OHCA and EMS systems-of-care, may have contributed to worse outcomes and should be considered in public health strategies.

## Figures and Tables

**Figure 1 ijerph-18-03646-f001:**
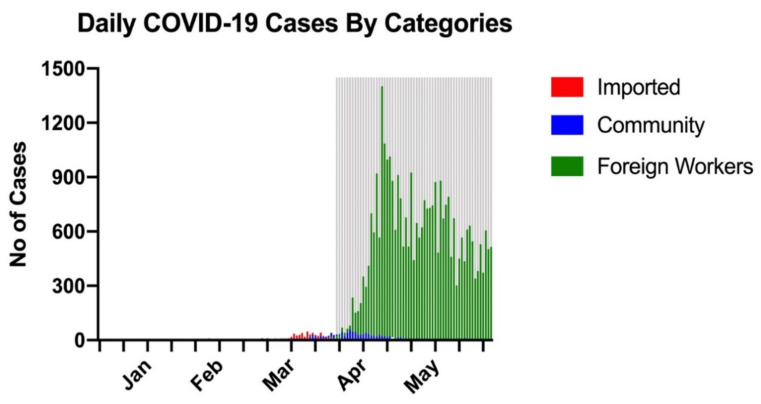
Daily COVID-19 cases by categories (imported, community and foreign workers) from January to May 2020. Area shaded in grey represents Circuit Breaker (3 April to 2 June 2020). Abbreviations: COVID-19, Coronavirus Disease 2019.

**Figure 2 ijerph-18-03646-f002:**
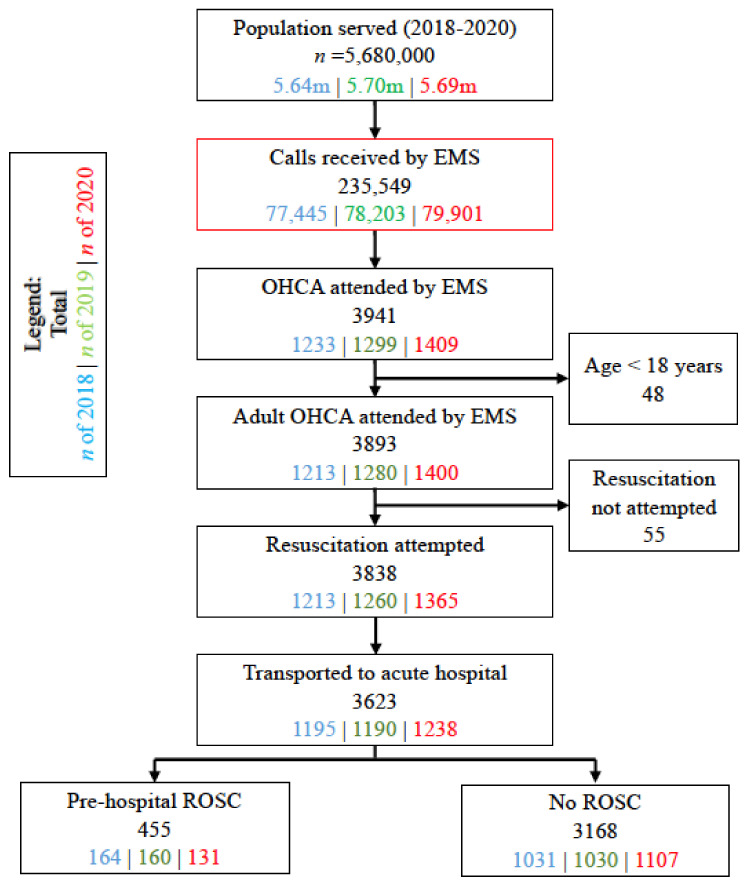
Patient selection between 1 Jan to 31 May 2018 to 2020. Abbreviations: m, million; EMS, Emergency Medical Services; OHCA, out-of-hospital cardiac arrest; ROSC, return of spontaneous circulation.

**Figure 3 ijerph-18-03646-f003:**
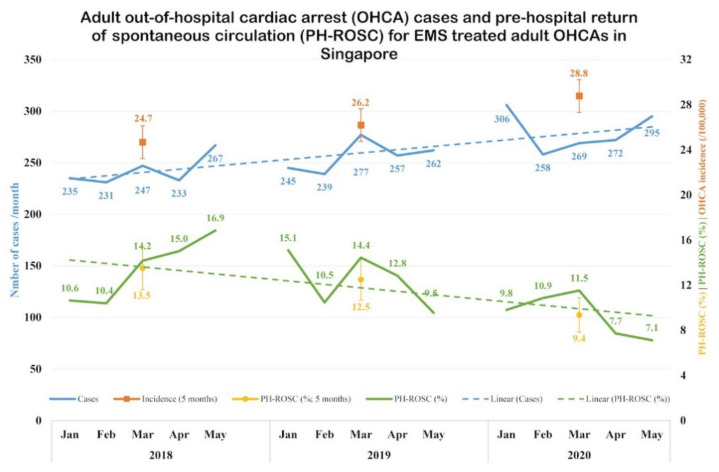
Temporal trends in incidence and outcome of EMS-attended OHCA. Temporal trends of OHCA incidence and outcome in Singapore from January to May 2018 to 2020, with pre-hospital ROSC as the outcome. Incidence is presented as number of cases and incidence rate (per 100,000 population). Outcome is presented as proportion (%) of pre-hospital ROSC. Abbreviations: EMS, Emergency Medical Services; OHCA, out-of-hospital cardiac arrest; ROSC, return of spontaneous circulation.

**Table 1 ijerph-18-03646-t001:** Baseline characteristics of adult EMS-attended OHCA patients.

Characteristics (%)	Jan–May 2018 (*n* = 1213)	Jan–May 2019 (*n* = 1280)	Jan–May 2020 (*n* = 1400)
Age in years, median (Q1–Q3)	71 (59–82)	71 (60–83)	73 (60–84)
Gender, male	779 (64.2)	818 (63.9)	882 (63.0)
Race			
Chinese	804 (66.3)	905 (70.7)	992 (70.9)
Malay	201 (16.6)	192 (15.0)	199 (14.2)
Indian	142 (11.7)	135 (10.6)	171 (12.2)
Others	66 (5.4)	48 (3.8)	38 (2.7)
Location Type			
Home residence	866 (71.4)	943 (73.7)	1081 (77.2)
Healthcare facilities	90 (7.4)	108 (8.4)	137 (9.8)
Public areas	229 (18.9)	196 (15.3)	161 (11.5)
In EMS/Private ambulance	25 (2.1)	25 (2.0)	17 (1.2)
Others	3 (0.3)	8 (0.6)	4 (0.3)
Arrest Witnessed By			
Not witnessed	563 (46.4)	690 (53.9)	533 (38.1)
EMS/Private ambulance	129 (10.6)	130 (10.2)	157 (11.2)
Bystander	521 (43.0)	460 (35.9)	710 (50.7)
Family member	295 (56.6)	241 (52.4)	464 (65.4)
Lay person	184 (35.3)	170 (37.0)	145 (20.4)
Healthcare provider	42 (8.1)	49 (10.7)	101 (14.2)
Bystander Interventions			
Bystander CPR performed			
No CPR	466 (38.4)	508 (39.7)	671 (47.9)
Unassisted CPR	227 (18.7)	238 (18.6)	272 (19.4)
DA-CPR	520 (42.9)	534 (41.7)	457 (32.6)
First CPR initiated by			
Family	475 (39.2)	432 (33.8)	438 (31.3)
Non-related layperson	272 (22.4)	340 (26.6)	291 (20.8)
Bystander AED applied	66 (5.4)	142 (11.1)	131 (9.4)
First arrest rhythm			
Shockable rhythm	191 (15.7)	198 (15.5)	206 (14.7)
Non-shockable rhythm	1014 (83.6)	1060 (82.8)	1178 (84.1)
Pre-hospital Defibrillation	313 (25.8)	296 (23.1)	298 (21.3)
Total response time in minute, median (Q1–Q3)	11.9 (9.8–14.9)	11.3 (9.4–13.5)	12.6 (10.5–15.1)
-Call received to dispatch	2.4 (1.8–3.2)	2.1 (1.6–2.9)	2.0 (1.5–2.8)
-Dispatch to scene arrival	6.1 (4.5–8.4)	6.0 (4.6–8.0)	6.3 (4.9–8.2)
-Scene arrival to first patient contact	3.2 (2.2–4.7)	3.0 (1.9–4.2)	3.8 (2.5–5.4)
* Scene time in minute, median (Q1–Q3)	22.4 (19.0–25.9)	22.7 (19.5–26.3)	24.3 (20.8–28.1)
EMS Outcome			
Resuscitation at scene	1213 (100.0)	1260 (98.4)	1365 (97.5)
Transported to acute hospital	1195 (98.5)	1190 (93.0)	1238 (88.4)
Pre-hospital ROSC	164 (13.5)	160 (12.5)	131 (9.4)

Numbers are *n* (%) for categorical variables and median (Q1–Q3) for continuous variables. * Data are missing for 17 cases in 2018, 89 cases in 2019 and 166 cases in 2020. Abbreviations: EMS, emergency medical services; OHCA, out-of-hospital cardiac arrest; Q1–Q3, first to third quartile; CPR, cardiopulmonary resuscitation; DA, dispatch-assisted; AED, automated external defibrillator; ROSC, return of spontaneous circulation.

**Table 2 ijerph-18-03646-t002:** Binary logistic regression comparing characteristics and management of OHCA between pandemic and pre-pandemic periods.

OHCA Characteristics (%)	Event vs. Reference Level	Pandemic vs. Pre-Pandemic	
		Unadjusted Odds Ratio (95% CI)	*p*-Value
Age	≥65 vs. <65	1.15 (1.00, 1.32)	0.052
Gender	Female vs. Male	1.05 (0.91, 1.20)	0.509
Race	Chinese vs. non-Chinese	1.12 (0.97, 1.29)	0.134
Location Type	Home residence vs. Public areas	1.28 (1.10, 1.49)	0.002
Witnessed arrest	Yes vs. No	1.64 (1.44, 1.88)	<0.001
First arrest rhythm	Shockable vs. Non-shockable	0.92 (0.77, 1.11)	0.384
Bystander CPR performed	Yes vs. No	0.70 (0.61, 0.80)	<0.001
Bystander AED applied	Yes vs. No	1.13 (0.90, 1.43)	0.282
Call received to dispatch	≥2 vs. <2	0.66 (0.58, 0.75)	<0.001
Dispatch to scene arrival	≥6 vs. <6	1.17 (1.02, 1.33)	0.022
Scene arrival to first patient contact	≥2 vs. <2	1.41 (1.20, 1.66)	<0.001
Total response time	≥10 vs. <10	1.71 (1.46, 2.00)	<0.001
Pre-hospital ROSC	Yes vs. No	0.69 (0.56, 0.86)	<0.001

Pandemic period: Jan to May 2020; Pre-pandemic period: Jan to May 2018, Jan to May 2019. Abbreviations: OHCA, out-of-hospital cardiac arrest; CPR, cardiopulmonary resuscitation; AED, automated external defibrillator; EMS, emergency medical services; ROSC, return of spontaneous circulation.

**Table 3 ijerph-18-03646-t003:** Multivariable logistic regression of OHCA event characteristics and outcomes between pandemic and pre-pandemic periods.

Variable	Event vs. Reference Level	Pandemic vs. Pre-Pandemic	
		Adjusted Odds Ratio (95% CI)	*p*-Value
*OHCA characteristics* ^1^			
Location Type	Home residence vs. Public areas	1.48 (1.24, 1.75)	<0.001
Witnessed arrest	Yes vs. No	1.71 (1.49, 1.97)	<0.001
Bystander CPR performed	Yes vs. No	0.70 (0.61, 0.81)	<0.001
Bystander AED applied	Yes vs. No	1.63 (1.26, 2.10)	<0.001
*Clinical Outcomes* ^2^			
Pre-hospital ROSC	Yes vs. No	0.67 (0.53, 0.84)	<0.001

Pandemic period: Jan to May 2020; Pre-pandemic period: Jan to May 2018, Jan to May 2019. ^1^ Multivariable logistic regression of OHCA characteristics, accounting for age (continuous), gender, first rhythm of arrest, location type, witnessed arrest and bystander interventions. Outcome is taken as the year, with reference year being the combined of 2018+2019. ^2^ Multivariable logistic regression of outcome, accounting for age (continuous), gender, location type, witnessed arrest, bystander interventions, first rhythm of arrest, pre-hospital defibrillation and total response time (continuous). Abbreviations: OHCA, out-of-hospital cardiac arrest; CPR, cardiopulmonary resuscitation; AED, automated external defibrillation; ROSC, return of spontaneous circulation.

## Data Availability

The data supporting the findings of this study are available from the corresponding author upon reasonable request, subject to approval by the local institution.

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
