# Peer review of "Impact of COVID-19 on Out-of-Hospital Cardiac Arrest in Singapore"

_ijerph, 2021, doi:10.3390/ijerph18073646_

Round 1
Reviewer 1 Report
I appreciate that you consider me a review of this article, which I find very interesting according to my line of research.
Introduction: The introduction provides sufficient background and contextualizes the topic of study
- Line 58: illness
- Line 75: this study cannot evaluate the impact of COVID-19 on OHCA incidence; This compares the incidence of OHCA during the COVID-19 pandemic period (as shown in the conclusion)
Methods:
- Line 81: STROBE guidelines should be cited
- Line 112: citation no. 17 is not relevant in this part of the manuscript, it should be deleted
- Is SCDF EMS data repository the only official OHCA registry in Singapore?
Statistical Analysis:
- It is not specified which variables were treated as categorical and which as continuous.
- Line 175: why was the significance level set at p <0.02 in the variable selection stage?
Results:
- Line 186: the number of OHCA cases (3893) in adults does not coincide with that of Figure 2
- Figure 3: this figure is difficult to read, it must be enlarged
Discussion:
- Why do you think dispatcher-assisted CPR is lower in 2020 if "Call received to dispatch" shows OR lower than previous periods? this aspect should be discussed
Congratulations on your work.
Reviewer 2 Report
The submission by Lim et al. is an observational study aimed to assess the incidence and outcomes of out-of-hospital cardiac arrest (OHCA) in Singapore during the Corona-virus disease 2019 (COVID-19) pandemic, compared with non-pandemic periods. This manuscript adds public health data similar to other studies from populated cities in France, Italy and the USA. Data reflects the cumulative incidence of out-of-hospital cardiac arrest in the year 2020 associated with the presence of COVID-19. Thus, the topic adds information to public health entities and governmental units to reconsider health strategies to avoid disruption and management of patients with acute cardiovascular needs.
The manuscript is well written and incorporated specific stats. The authors commented briefly on the potential biases of the study. I believe the manuscript would benefit from further elaboration/clarification of the reported biases and confounding factors.
Reviewer 3 Report
Overall, the authors have submitted an interesting paper that fits to the aim of the journal. I have a few comments that authors should address before the paper can be published.
- I am a bit astonished that the number of OHCA seen by EMS has increased during COVID, as some countries as e.g. Germany have experienced a decrease, leading to more patients to be seen by doctors later than before (often leading to worse outcomes). Do you know if numbers of critically-injured patients for the EMS also stayed the same overall or even also increased during the pandemic?
- Abstract: why is this a preliminary study? Not the most ideal beginning for a journal paper
- Abstract: lines 32/33 (“(median age—first-third 32 quartile—72—60–83—years, 63.7% males)”) are difficult to understand
- Page 2, line 82: reference missing for the guidelines
- Page 5, Figure 2: I think there is some mistake in the numbers as they do not add up for all steps: potentially the “Adult OHCA attended by EMS” is not correct, but it might also be around that
- Page 7, Figure 3: could be sharper
- Did you have or will you be able to get hospital data about the actual patient outcomes?
- Page 9: did the total number of ambulance services also increase or only calls? If only calls, why did the response time increase exactly? Was it basically the chute time or did also driving times increase? If yes, why?
- Accordingly, can you add some general numbers for the EMS service to better understand the numbers and results?
